# Development of Multilayer Nanoparticles for the Delivery of Peptide-Based Subunit Vaccine against Group A *Streptococcus*

**DOI:** 10.3390/pharmaceutics14102151

**Published:** 2022-10-10

**Authors:** Jolynn Kiong, Ummey Jannatun Nahar, Shengbin Jin, Ahmed O. Shalash, Jiahui Zhang, Prashamsa Koirala, Zeinab G. Khalil, Robert J. Capon, Mariusz Skwarczynski, Istvan Toth, Waleed M. Hussein

**Affiliations:** 1School of Chemistry and Molecular Biosciences, The University of Queensland, St Lucia, Brisbane, QLD 4072, Australia; 2Institute for Molecular Bioscience, The University of Queensland, St Lucia, Brisbane, QLD 4072, Australia; 3School of Pharmacy, The University of Queensland, Woolloongabba, Brisbane, QLD 4072, Australia

**Keywords:** peptide vaccine, delivery systems, polyelectrolyte complexes, cationic polymers, polylysine, Group A *Streptococcus*

## Abstract

Peptide-based subunit vaccines include only minimal antigenic determinants, and, therefore, are less likely to induce allergic immune responses and adverse effects compared to traditional vaccines. However, peptides are weakly immunogenic and susceptible to enzymatic degradation when administered on their own. Hence, we designed polyelectrolyte complex (PEC)-based delivery systems to protect peptide antigens from degradation and improve immunogenicity. Lipopeptide (LCP-1) bearing J8 B-cell epitope derived from Group A *Streptococcus* (GAS) M-protein was selected as the model peptide antigen. In the pilot study, LCP-1 incorporated in alginate/cross-linked polyarginine-J8-based PEC induced high J8-specific IgG antibody titres. The PEC system was then further modified to improve its immune stimulating capability. Of the formulations tested, **PEC-4**, bearing LCP-1, alginate and cross-linked polylysine, induced the highest antibody titres in BALB/c mice following subcutaneous immunisation. The antibodies produced were more opsonic than those induced by mice immunised with other PECs, and as opsonic as those induced by antigen adjuvanted with powerful complete Freund’s adjuvant.

## 1. Introduction

Vaccination is one of the most effective medical interventions against infectious diseases [1]. Attenuated and inactivated vaccines have been on the market for over two centuries. However, they can present a variety of adverse effects, as they carry a wide range of materials that do not contribute to achieving required immune responses [2]. To avoid such drawbacks, peptide-based subunit vaccines carry only minimal antigenic determinants (epitopes) required to induce desired immune responses [3]. However, peptide epitopes are weakly immunogenic and susceptible to enzymatic degradation [4]. Thus, co-administration of peptide epitopes with adjuvants/delivery systems to deliver them into antigen presenting cells (APCs) for further processing and presentation to B- and T-cells is required [3,5]. A limited number of adjuvants have been approved for human use. These are generally safe; however, they may also cause undesired immune responses, e.g., excessive inflammation [6]. Instead of coadministration with an adjuvant, peptide antigens can also be conjugated to carriers, such as lipids, to improve their immunogenicity. For example, lipid core peptide (LCP) was able to induce strong humoral immune responses in conjunction with a variety of antigens [7,8,9]. Moreover, increasing hydrophobicity via inclusion of lipid moieties allows vaccines to self-assemble into nanoparticles, preventing premature enzymatic degradation and increasing uptake by APCs [10].

Group A *Streptococcus* (GAS) is a highly prevalent Gram-positive bacterium, transmitted through contaminated food and/or respiratory droplets from an infected host [11]. It predominantly colonizes the throat and the skin, causing a broad spectrum of diseases, ranging from superficial to severe, invasive infections [12]. Though antibiotics are still effective against GAS infection, antibiotic-resistant GAS strains have emerged [13]. Rapid disease recurrence after treatment and delayed or ineffective medical intervention often results in deadly post-infectious complications, such as rheumatic heart disease (RHD). The high infection rate of GAS and limited success with treatments have triggered strong interest in the development of a safe and effective vaccine [13,14].

For decades, GAS vaccine development focused on the use of protein-based antigens, especially the major virulent factor of GAS: M-protein. However, the use of the entire M-protein as an antigen was discontinued due to the potential for autoimmune response [15], as recorded in early clinical trials [16]. Furthermore, several studies found that M-protein contains B- and T-cell epitopes cross-reactive with human tissues [17,18,19]. As a result, non-cross-reactive M-protein peptide segments have been used as a viable alternative for GAS vaccine development [20]. An M-protein-derived B-cell epitope, J8 (QAEDKVKQSREAKKQVEKALKQLEDKVQ), was exploited in the development of a peptide-based GAS vaccine [21] that recently completed phase I clinical trials [22]. In addition, LCP systems have been widely used in GAS vaccine research [7]. LCP is built from lipoamino acids (e.g., 2-amino-d,l-hexadecenoic acid), a branching moiety, such as lysine, and peptide epitopes (Figure 1A). Toll-like receptor-2 (TLR-2) on dendritic cells can recognize lipoamino acids, making them self-adjuvanting moieties, while the conjugation of peptides to lipids can protect peptides from enzymatic degradation [7].

Polyelectrolyte complexes (PECs) are formed through electrostatic interactions between oppositely charged polymeric macromolecules [23]. Polymers, such as cationic chitosan, polyethyleneimine (PEI) and anionic alginate, have been widely utilized in PEC-based drug delivery systems [24]. Moreover, nanoparticles built from PECs often have strong adjuvanting properties [25]. PEI, one such polymer-based vaccine delivery system, is made up of primary, secondary and tertiary amine groups connected via ethylene linkers [26]. PEI has the ability to increase the uptake of entrapped antigen by APCs [27]. When co-formulated with alginate, PEI can allow controlled release of loaded antigens, prolonging their exposure to immune cells [28]. However, PEI is non-biodegradable, non-biocompatible, toxic and not easily derivatized [29]. To overcome these limitations, we designed disulphide cross-linked polycationic peptide-based nanoparticles. These nanoparticles were then modified to carry mannose as an APC targeting moiety.

Here, LCP-1, composed of J8 B-cell epitope, P25, a universal T-helper epitope, and LAA-C16 (2-amino-d,l-hexadecanoic acid), was incorporated into PECs (Figure 1). LCP-1 was complexed with alginate, then coated with one of the cross-linked cationic peptides: polyArg-J8 (Ac-CRRRCRSRRC-NH_2_ cross-linked by disulphide bond with Ac-CQAEDKVKQSREAKKQVEKALKQLEDKVQ-NH_2_); cross-linked poly-Arg (Ac-CRRRCRSRRC-NH_2_); mannosylated poly-Arg (Ac-mannose-SGGGGGGCRRRCRRRC-NH_2_); or poly-Lys (Ac-CKKKCKSKKC-NH_2_), to form **PEC-1–PEC-4**, respectively (Figure 1). Prior to coating, peptide cross-linking was achieved through oxidation of the cysteine moieties. The control formulation, **PEC-0**, was formulated in the same manner, by complexing LCP-1 with alginate, followed by coating with PEI 25 kDa [30]. The immune simulating ability of **PEC-1** and **PEC-0** was tested first in a pilot study with mice. **PEC-1–PEC-4**, along with uncoated cross-linked polyArg-J8, positive control: P25-J8/complete Freund’s adjuvant (CFA), and negative control: PBS were then evaluated for their ability to stimulate antibody production. The opsonic capacity of the produced antibodies was tested against two clinically isolated GAS strains: D3840 and GC2203.

## 2. Materials and Methods

### 2.1. Materials

All reagents were analytical grade or equivalent, unless otherwise stated. Protected Fmoc amino acids were purchased from Novabiochem (Läufelfingen, Switzerland). Hexafluorophosphate azabenzotriazole tetramethyl uronium (HATU) was obtained from Mimotopes (Melbourne, VIC, Australia). Triisopropylsilane (TIPS) was purchased from Thermo Scientific (Scoresby, VIC, Australia). *N*, *N*-diisopropylethylamine (DIPEA), methanol (MeOH), dichloromethane (DCM), acetonitrile, *N*, *N*-dimethylformamide (DMF), trifluoroacetic acid (TFA), diethyl ether, 37% hydrochloric acid, piperidine and skim milk (microbiology grade) were purchased from Merck (Darmstadt, Germany). Rink amide 4-methylbenzhydrylamine (MBHA) resin was purchased from Novabiochem (Hohenbrunn, Germany). O-phenylenediamine dihydrochloride (OPD) tablets were obtained from SIGMAFAST^TM^—Sigma-Aldrich (Sydney, NSW, Australia). Phosphate-buffered saline (PBS) tablets were purchased from Gibco (Paisley, UK). Goat anti-mouse IgG-horseradish peroxidase (HRP) was purchased from Bio-Rad (Hercules, CA, USA). Goat anti-mouse IgG1- and IgG2a-HRP conjugated secondary antibodies were purchased from Invitrogen (Waltham, MA, USA). Ammonium bicarbonate, sodium alginate and Tween 20 were purchased from Sigma-Aldrich (Sydney, NSW, Australia).

### 2.2. Instruments

Analytical analysis was performed on a Shimadzu LCMS-2020 analytical reverse-phase high-performance liquid chromatography (RP-HPLC) instrument (Kyoto, Japan), using a Vydac analytical C-18 column (218TP; 10 μm, 250 mm × 4.6 mm) with a flow rate of 1 mL/min. Peptide purification was performed on a preparative Shimadzu RP-HPLC instrument, using an Altima preparative C-18 column (218TP; 10 mm, 250 mm × 22 mm) with a flow rate of 20 mL/min. Both detections were performed at 214 nm.

Transmission electron microscopy (TEM; HT7700 Exalens, HITACHI Ltd., Tokyo, Japan) was performed at the Australian Microscopy & Microanalysis Research Facility, Centre for Microscopy and Microanalysis, The University of Queensland (UQ).

### 2.3. Peptide Synthesis & Purification

Peptides of interest were synthesized using fluorenylmethoxycarbonyl solid-phase peptide synthesis (Fmoc-SPPS), following a reported protocol [31] with some modifications. Rink amide MBHA (0.1 mmol) was swollen in DMF for at least 2 h before synthesis. A short treatment with 20% piperidine in DMF was completed prior to each amino acid coupling for Fmoc deprotection (5 min and 10 min). The amino acids (4.2 equivalent) were activated by dissolution in 0.5 M HATU in DMF (4.0 equivalent), then in 6.2 equivalent of DIPEA, where each amino acid was coupled twice (2 × 20 min). After coupling the first amino acid of the sequence, acetylation was performed to act as a cap for unreacted amino groups on the resin. Once all amino acids were coupled, acetylation was performed again prior to peptide cleavage.

In preparation for cleavage, resins were washed with DMF, DCM and MeOH, then stored in the desiccator overnight to dry. Once dried, the peptide sequence was cleaved off from the resin using TFA cocktail, TFA:TIPS:MilliQ water (95%:2.5%:2.5%) [31]. The peptides were verified and assessed through analytical RP-HPLC with a C18 column and mass spectrometry (MS). Purification of the peptides was performed through preparative RP-HPLC-C18.

**Poly-Arg** (Ac-CRRRCRSRRC-NH_2_): HPLC (C18): t*_R_* = 15.3 min, purity ≥ 98%. Mol. Wt.: 1392.70 g/mol. ESI-MS: *m*/*z* [M+1H]^1+^: 1393.8 (calculated 1393.7), [M+2H]^2+^: 697.0 (calculated 697.4), [M+3H]^3+^: 465.6 (calculated 465.2); yield: 38%.

**J8-Cys**: HPLC (C18): t*_R_* = 13.9 min, purity ≥ 98%. Mol. Wt.: 3426.95 g/mol. ESI-MS: *m*/*z* [M+3H]^3+^: 1143.2 (calculated 1143.3), [M+4H]^4+^: 857.7 (calculated 857.7), [M+5H]^5+^: 686.3 (calculated 686.4), [M+6H]^6+^: 572.6 (calculated 572.2); yield: 56%.

**α- and β-mannosylated poly-Arg** (Ac-mannose-SGGGGGGCRRRCRRRC-NH_2_): HPLC (C18): t*_R_* = 7.9 and 8.4 min, purity ≥ 99%. Mol. Wt.: 1897.87 g/mol. ESI-MS: *m*/*z* [M+2H]^2+^: 948.7 (calculated 949.6), [M+3H]^3+^: 633.5 (calculated 633.4), [M+4H]^4+^: 475.3 (calculated 475.3); yield: 60%.

**Poly-Lys** (Ac-CKKKCKSKKC-NH_2_): HPLC (C18): t*_R_* = 13.0 min, purity ≥ 99%. Mol. Wt.: 1224.80 g/mol. ESI-MS: *m*/*z* [M+1H]^1+^: 1224.9 (calculated 1225.8), [M+2H]^2+^: 613.2 (calculated 613.4), [M+3H]^3+^: 409.3 (calculated 409.3); yield: 31%.

### 2.4. Oxidation

Pure peptides were oxidized using a general method of disulphide cross-linking. Oxidation was performed on 5 mg of each poly-cationic peptide. However, a mixture of J8-Cys (4.3 mg) and poly-Arg (2.1 mg) was used for oxidation in the synthesis of J8-Cys-poly-Arg. Peptides were dissolved in a freshly prepared 50 mM ammonium bicarbonate solution (200 μL) and stirred overnight, freeze-dried, then 0.1 M hydrochloric acid (0.5 mL) was added. They were then freeze-dried again. To test the success of the disulphide cross-linking reaction, Ellman’s test was performed to determine the presence of free thiols, according to Thermo Scientific’s user guide for Ellman’s reagent [32]. Cysteine standard solution was prepared to quantify free thiols in the samples.

### 2.5. Formulation and Optimization of LCP-1/Alginate

A stock solution (4 mg/mL) of alginate was prepared in MilliQ water, swollen for 24 h at room temperature and stored at 4 °C. Subsequently, LCP-1 (1 mg) was dissolved in MilliQ water (1 mL). Swollen alginate solutions (50 μL, each) (containing 20, 30, 40, 50 or 60 μg) were added dropwise into LCP-1 solutions (100 μg each), at a rate 3 μL over 15 s, with 1 min intervals between each addition. The mixtures were sonicated in an ice bath twice, for 2 min, each, using a probe-type sonicator (Ultrasonic Homogenizer Model 3000, Biologics, Inc., Cary, NC, USA) at 120 W. The mixtures were then stirred at room temperature for 1 h. Particle size, zeta potential and polydispersity index (PDI) were detected using dynamic light scattering (DLS) (Malvern Instrument, Malvern, UK).

### 2.6. Optimization of LCP-1/Alginate/Cross-Linked Poly-Cationic Peptide

After selecting the best formulation of LCP-1/alginate (0.1 mg LCP-1 to 40 μg alginate), 50 μL of cross-linked poly-cationic peptide (containing either 30, 40, 50, 60, 70, 80, 90 or 100 μg) was added dropwise at a rate of 5 μL over at least 15 s into LCP-1/alginate complex (100 μg of LCP-1). The mixture was stirred for 1 h, then size and charge were assessed via DLS to determine the best concentration for each complex.

### 2.7. Transmission Electron Microscopy

TEM was used to visualize surface morphology. Each sample was dissolved in PBS, then one drop was added to a glow-discharged carbon-coated grid and allowed to settle for 2 min. Excess liquid was wicked off with filter paper. Particles were then stained with 2% uranyl acetate (pH 7) for 2 min. The excess stain solution was wicked away and the grid was air-dried for 5 min before particle imaging.

### 2.8. Encapsulation Study

To evaluate encapsulation efficiency (EE%), ultracentrifugation of the formulations was employed. A volume of 0.25 mL of each formulation was transferred into a 1 mL polycarbonate thick-wall ultracentrifuge tube. The formulations were further diluted with another 0.25 mL of PBS, then mixed by pipetting up and down for 1 min. The tubes were placed in an ultracentrifuge (Optima™ MAX-XP, Beckman Coulter, Brea, CA, USA) equipped with a TLA120.2 rotor. Sample centrifugation was conducted at 100,000 RPMs for 1 h at 4 °C. LCP-1 concentration was determined as follows: a volume of 100 µL of the supernatant was freeze-dried, redissolved in 0.5 mL of 50% solvent B, and 100 µL of each redissolved sample, as well as a similar volume of standard (LCP-1) lipopeptide solution (1 mg/mL) in solvent A (water + 0.1% TFA) and solvent B (90% Acetonitrile + 10% water + 0.1% TFA) (1:1). These were then put through RP-HPLC (Shimadzu Corp., Kyoto, Japan) using a time gradient program (0 to 100% solvent B, 40 min; C4 column). In parallel, 60 µL of each fresh formulation was diluted with 60 µL of PBS, and subjected to HPLC following the same method as a reference concentration. EE% values were calculated using Equation (1).
(1)$$\mathrm{EE}\%=\left[1-\frac{\mathrm{Concentration}\;\left(\mathrm{found}\right)}{\mathrm{Concentration}\;\left(\mathrm{reference}\right)}\right]\times 100$$

Equation (1): Formula to determine encapsulation efficiency.

### 2.9. Animal Studies

All animal protocols were approved by The University of Queensland Animal Ethics Committee (SCMB/AIBN/069/17). Two animal studies were performed: both used the same number of mice and vaccine administration method.

BALB/c mice (6–8 weeks old; 60 in total: 4 groups in the first, 8 groups in the second) were divided randomly into sample groups (n = 5). The mice were inoculated subcutaneously with freshly prepared vaccine candidates equivalent to 30 μg of LCP-1 on days 0, 25 and 50. Blood was collected via tail bleed 10 days after each immunisation. A final blood collection was done following euthanasia on day 64 via heart puncture. Tail blood samples (20 μL per mouse) were mixed with 180 μL of 1X PBS. The serum was separated via centrifugation at 3600 RPMs for 10 min. All biological samples were stored at −80 °C prior to further analysis.

### 2.10. Determination of Antibody Titres & Statistical Analyses

J8-specific IgG antibody titres from sera were measured via enzyme-linked immunosorbent assay (ELISA), following a published protocol [33]. First, 96-well microplates were coated with J8 (50 μg/plate) in carbonate coating buffer (pH 9.6) and incubated at 37 °C for 90 min. Blocking was done with 5% (*w*/*v*) skim milk in PBS overnight at 4 °C. Serum samples were added to the first row of the plate, starting with 1/100 dilution, then serially diluted two-fold down the plates. The plates were then incubated at 37 °C for 90 min. HRP-conjugated goat anti-mouse secondary IgG antibody was added to each well, and the plates were incubated again under the same conditions. OPD substrate was added, and the plates were incubated in dark conditions for 20 min before absorbance measurements were taken at 450 nm (SpectraMax microplate reader, Molecular Devices, San Jose, CA, USA). Endpoint antibody titres were determined as the highest dilution, for which the absorbance value was > 0.065, 3 standard deviations above naïve mean absorbance and where the difference between previous three dilution values was > 0.005. Statistical significance was analysed using one-way ANOVA, followed by Tukey’s test for comparison between groups with GraphPad Prism 9.0.

### 2.11. Opsonization Assay of Antibodies against Group A Streptococcus

Antibody opsonization assays were performed with GAS clinical isolate strains provided by the Princess Alexandra Hospital: strains GC203 (wound swab) and D3840 (nasopharynx swab) [34]. The two strains were streaked on Todd Hewitt broth (THB) agar plates with 5% yeast supplementation and incubated at 37 °C for 24 h. Single colonies from the plates were inoculated in 5 mL of THB + 5% yeast and incubated until bacterial replication reached approximately 4.6 × 10^6^ colony forming units (CFU)/mL. Sera collected from the heart (day 64) was heat-inactivated at 50 °C for 30 min. The sera were then mixed with a bacterial aliquot and horse blood at a ratio of 1:1:8, respectively. The mixtures were added to 96-well plates containing a 2-fold dilution of bacterial culture. The plates were incubated at 37 °C for 3 h. Then, 10 μL aliquots from the wells were streaked onto plates with THB agar + 5% yeast and 5% horse blood, and incubated for another 24 h [34]. Bacterial survival rates were calculated based on the number of colonies on the plate. Opsonic activity was determined by the percent reduction in average CFU, following the formula below (Equation (2)) [35].
(2)$$\left(1-\frac{\mathrm{CFU}\;\mathrm{in}\;\mathrm{presence}\;\mathrm{of}\;\mathrm{serum}}{\mathrm{average}\;\mathrm{CFU}\;\mathrm{in}\;\mathrm{presence}\;\mathrm{of}\;\mathrm{media}}\right)\times 100\%$$

Equation (2): Formula for quantifying antibody opsonic activity.

## 3. Results and Discussion

### 3.1. Design of the Vaccine Delivery System

PEI is a highly cationic polymer that has been widely used in the development of vaccine delivery systems due to its adjuvanting capacity [36]. However, it is non-biodegradable and highly toxic. Here, we assessed simple, non-toxic, polycationic molecules as a replacement for PEI, through fully chemically synthesised poly-cationic coating over LCP-1.

LCP-1, a self-adjuvanting lipopeptide, was complexed with negatively charged polysaccharide alginate to produce anionic particles. These particles were then coated with biodegradable cationic cross-linked polymers, namely poly-Arg-J8, poly-Arg, mannosylated poly-Arg and poly-Lys, to produce polyelectrolyte complexes **PEC-1** to **PEC-4**, respectively. Based on previous studies, many formulations including poly-Arg and poly-Lys were used safely in vivo without showing any level of toxicity [9,37,38,39].

All PECs (Table 1) were self-assembled into nanoparticles in water, through dropwise addition of anionic alginate into cationic LCP-1. Corresponding cationic cross-linked polymers were then added dropwise to coat the nanoparticles. Poly-Arg was selected as a replacement for PEI due to its capacity to act as a cell-penetrating peptide [40]. Poly-Lys was selected as a closer analogue of PEI due to its amine group (instead of guanidine in arginine). However, Poly-Lys was selected as a negative control as it has been reported to be an ineffective immune stimulator [41]. The inclusion of a targeting moiety (mannose in **PEC-3**) on the nanoparticle surface was hypothesized to enhance PEC immunogenicity due to increased interaction with APCs. Finally, J8 B-cell epitope was conjugated to poly-Arg (via disulphide bridge formation) to allow easy recognition of **PEC-1** by B-cells. The three polycationic peptides mentioned above, poly-Lys, poly-Arg and mannosylated poly-Arg, were designed to carry three cysteine residues each, while J8 possessed a single cysteine moiety to allow intermolecular oxidative cross-linking and, thus, formation of large polymeric structures mimicking the high molecular weight of PEI. Cross-linking was performed prior to coating of the LCP-1/alginate nanoparticles.

### 3.2. Formation and Characterization of PECs

Peptide polymers were synthesised using classical Fmoc-SPPS (Appendix A) and oxidatively cross-linked using 50 mM ammonium bicarbonate solution overnight before acidification with 0.1 M hydrochloric acid. Free thiol content from unreacted cysteine residue was determined by Ellman’s test: no free thiol was detected, reflecting completion of the oxidation process and successful cross-linking between peptide monomers through disulphide linkages. Functionalisation of poly-Arg with GAS B-cell epitope, J8, required synthesis of poly-Arg-J8 via disulphide bond formation between poly-Arg and J8-Cys peptide; this was performed following the same oxidation procedure.

Nanoparticle size was measured by DLS and confirmed by TEM. **PEC-1–PEC-4** formed similarly large nanoparticles (200–300 nm) with relatively low PDI (<0.2), except for **PEC-3**, which formed more polydisperse particles (PDI = 0.3). All PECs were positively charged (24–32 mV) (Table 1, Appendix A). In addition, the encapsulation efficiency of LCP-1 in both **PEC-2** and **PEC-4** formulations were determined with 99% and 93%, respectively.

### 3.3. Immunogenicity of PECs

Two animal studies were performed to determinate nanocomplex immunogenicity. The pilot study was designed to establish when a cross-linked cationic peptide can be used as a replacement for PEI. **PEC-1** was produced as LCP-1, complexed with alginate, coated with a fully chemically synthesised polycationic layer and cross-linked to poly-Arg-J8. **PEC-0**, bearing PEI 25 kDa, was used as a reference. BALB/c mice were immunised with **PEC-0**, **PEC-1**, P25-J8/CFA as a positive control, and PBS as a negative control. **PEC-1** induced similar humoral immune responses to **PEC-0** (containing PEI; Figure 2A), proving that toxic PEI can be replaced with cross-linked polycationic peptides. However, neither of these formulations were as effective at triggering antibody responses as CFA-adjuvanted antigen. Thus, to further probe the ability of cationic peptides to enhance immune responses against an antigen, a second in vivo animal study was carried out. Three other complexes (**PEC-2** to **PEC-4**) were examined in comparison to **PEC-1**; namely, **PEC-2**, bearing cross-linked poly-Arg (instead of cross-linked poly-Arg-J8 in **PEC-1**); **PEC-3**, carrying cross-linked mannosylated poly-Arg; and **PEC-4**, carrying cross-linked poly-Lys.

All PEC constructs induced antigen-specific immune responses at similar levels. IgG titres induced by **PEC-4** were not significantly different to those induced by CFA-adjuvanted P25-J8 (Figure 2B). The other PECs were less efficient than P25-J8/CFA in stimulating J8-specific antibody production. It was notable that **PEC-1** was able to trigger slightly higher IgG titres than **PEC-2** and **PEC-3**. This can be attributed to the presence of J8 epitope on the surface of **PEC-1**, promoting prompt recognition by B-cell receptors. Higher antigen dosage was also likely a factor (both LCP-1 and the external coating carried antigen). However, the differences were not statistically significant. Despite caring mannose targeting moiety, **PEC-3** did not induce higher IgG production than other vaccine candidates. Uncoated LCP-1/alginate anionic nanoparticles were also effective in triggering IgG production; however, to a lesser extent than **PEC-1** and **PEC-4**. All PECs and controls induced higher production of IgG1 than IgG2a, suggesting that the vaccine candidates induced predominantly Th2-biased immune responses (Appendix A).

### 3.4. Antibody Opsonic Capability

Antibodies produced in the sera by immunised mice were used to determine opsonic activity against two strains of GAS clinical isolates (Figure 3): D3840 and GC2203, isolated from a nasopharynx swab and wound swab, respectively. Opsonic abilities were calculated based on the reduction of CFU compared to sera from negative control mice, determining the quality and effectiveness of the antibodies. Antibodies produced by mice immunised with **PEC-4** had comparable opsonic activity to the positive control group (P25-J8/CFA), while those of all other mice were significantly less opsonic. **PEC-1** to **PEC-3** sera had similar levels of opsonic activity, despite the higher overall dosage of antigen introduced by **PEC-1**. Thus, cross-linked poly-Lys (**PEC-4**) was better able to stimulate the immune system and produced higher opsonic antibody titres compared to poly-Arg (**PEC-1 to PEC-3**). Interestingly, simple poly-Arg (not cross-linked) was previously shown to be effective in inducing humoral immunity [42], while poly-Lys lacked immune stimulating ability [41]. The higher efficacy of poly-Lys compared to poly-Arg in the current study may be related to non-specific uptake of poly-Arg (as a cell penetrating peptide) by non-immune cells [40]. Alternatively, cross-linking of poly-Lys could have altered its immune stimulating potency.

## 4. Conclusions

Nanoparticles have been widely explored for the delivery of vaccines where encapsulation of antigens with commercially available cationic polymers, such as chitosan or PEI, is utilized. However, further modifications, such as the incorporation of targeting moieties to enhance immune responses, can be challenging. In this study, we demonstrated that a fully chemically synthesized polycationic cross-linked poly-Lys peptide, but not poly-Arg, can have similar capacity as CFA in stimulating humoral immunity. While there was no significant difference in the ability to trigger opsonic antibody production between the PECs that were derivatised with mannose, B-cell epitope, or neither, having flexibility in derivatisation is advantageous for developing delivery systems. Importantly, this study showed that cross-linked polycationic peptides can be used to replace PEI for the development of safe vaccine delivery systems.

## Figures and Tables

**Figure 1 pharmaceutics-14-02151-f001:**
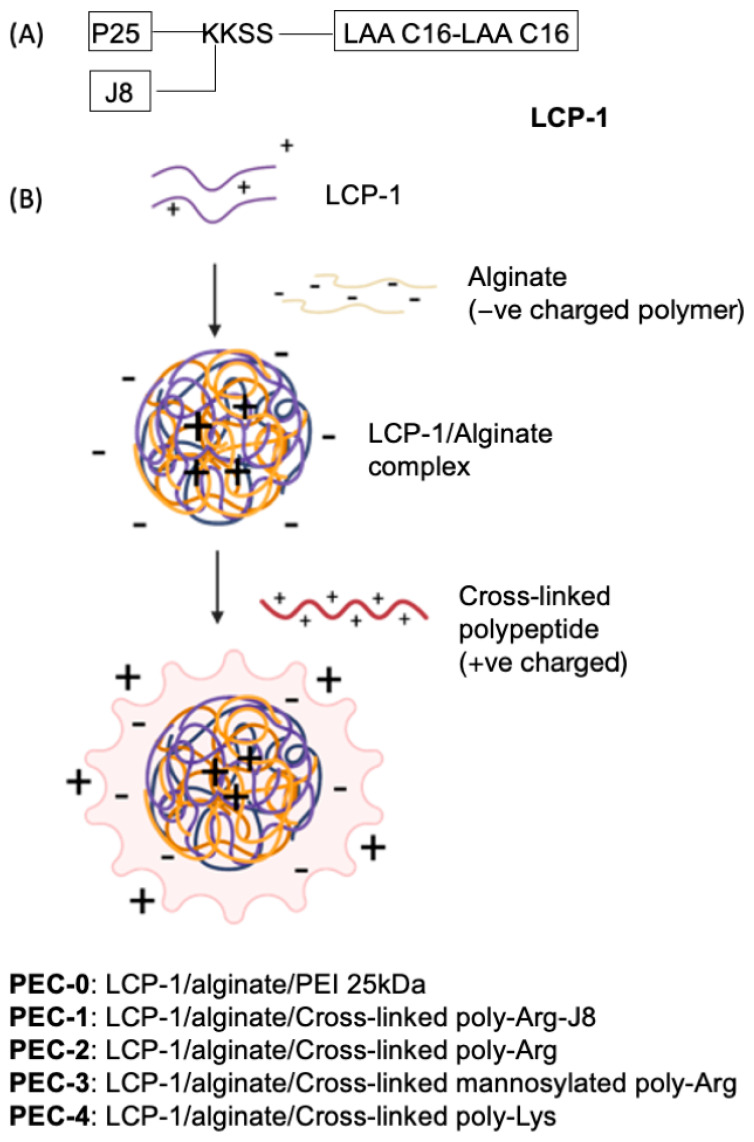
(**A**) The structure of **LCP-1**, bearing universal T-helper epitope, P25 (KLIPNASLIENCTKAEL), and B-cell epitope, J8 (QAEDKVKQSREAKKQVEKALKQLEDKVQ), derived from GAS M-protein. LAA C16: lipoamino acid, 2-amino-d,l-hexadecenoic acid; K: Lysine; S: Serine. (**B**) Schematic illustration of PEC formation through electrostatic attraction between differently charged layers, where LCP-1 is coated with polyanionic alginate then further coated with polycationic peptide.

**Figure 2 pharmaceutics-14-02151-f002:**
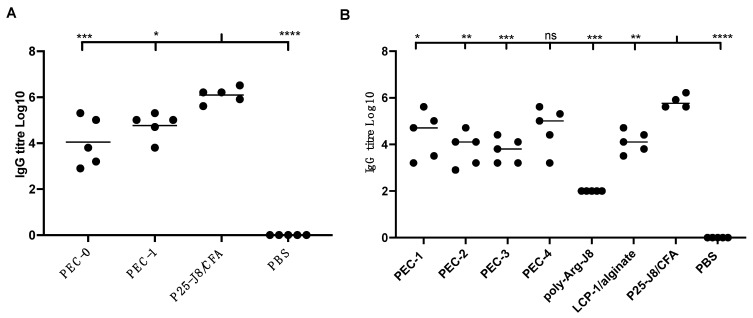
J8-specific IgG antibody titres in female BALB/c mice (n = 5) following subcutaneous immunisation with vaccine candidates and controls. (**A**) Results from the first animal study, with **PEC-0**, **PEC-1**, P25-J8/CFA and PBS. (**B**) Results from the second animal study, comparing **PEC-1** to **PEC-4** and controls that included poly-Arg-J8, LCP-1/alginate and P25-J8/CFA. Bars represent the average antigen-specific antibody titre. (ns, *p* > 0.05; *, *p* < 0.05; **, *p* < 0.01; ***, *p* < 0.001; ****, *p* < 0.0001 compared to P25-J8/CFA).

**Figure 3 pharmaceutics-14-02151-f003:**
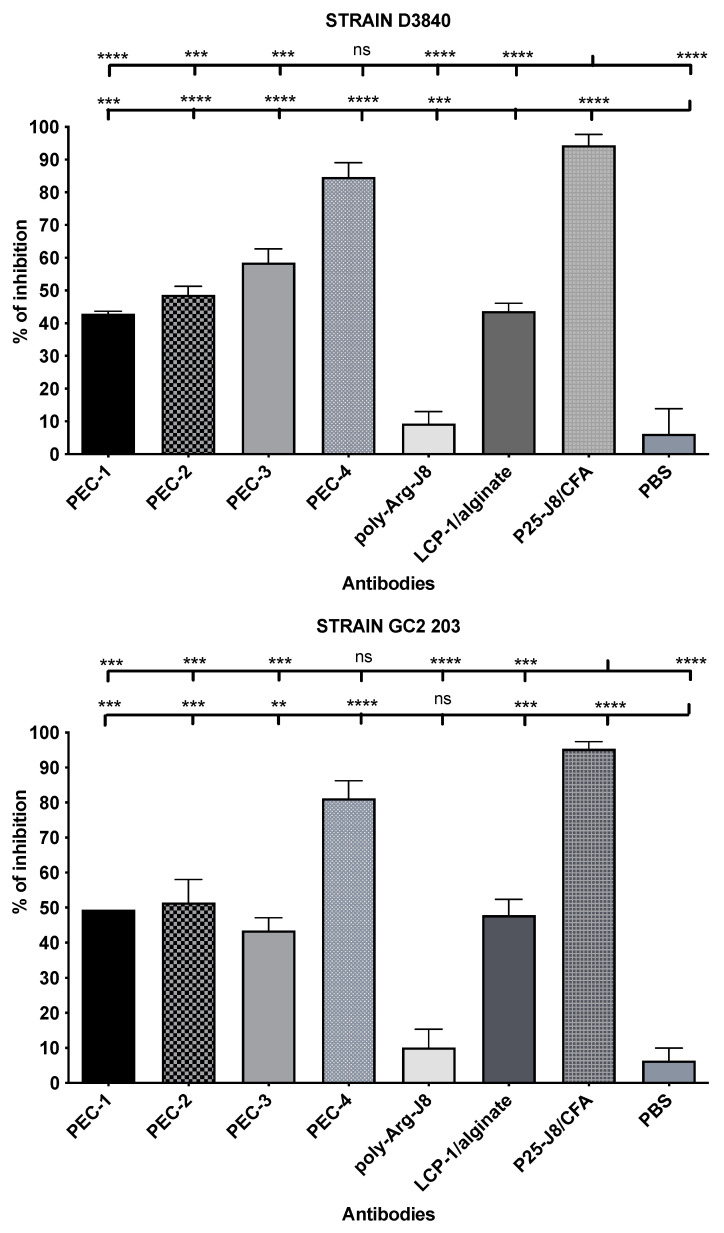
Opsonic activity (average opsonization %) of antibodies in mouse sera on day 64 post primary immunisation against GAS clinical isolates D3840 (nasopharynx swab) and GC2203 (wound swab). Error bars are shown for each group. For comparison, the group of interest is represented by upward bars (ns, *p* > 0.05; **, *p* < 0.01; ***, *p* < 0.001; ****, *p* < 0.0001).

**Table 1 pharmaceutics-14-02151-t001:** Physicochemical characterization of PECs.

Formulation	Negative Polymer	Cationic Polymer	Size (nm)	PDI	Zeta Potential (mV)	Initial Mass Ratio ^1^
**LCP-1/alginate**	alginate	----	176 ± 12	0.07 ± 0.02	−24 ± 1	10:4:0
**PEC-0**		PEI 25 kDa	153 ± 10	0.04 ± 0.02	33 ± 1	10:4:2
**PEC-1**		Cross-linked poly-Arg-J8	330 ± 56	0.19 ± 0.02	24 ± 1	10:4:11
**PEC-2**		Cross-linked poly-Arg	217 ± 1	0.10 ± 0.01	27 ± 1	10:4:7
**PEC-3**		Cross-linked mannosylated poly-Arg	286 ± 10 4300 ± 20	0.34 ± 0.04	32 ± 1	10:4:9
**PEC-4**		Cross-linked poly-Lys	191 ± 16 ~4000	0.17 ± 0.05	31 ± 1	10:4:6

PDI: polydispersity index. ^1^ Initial mass ratio = LCP-1:alginate:cationic polymer.

## Data Availability

The data presented in this study are available in this article.

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
