# Peer review of "Development of Multilayer Nanoparticles for the Delivery of Peptide-Based Subunit Vaccine against Group A Streptococcus"

_pharmaceutics, 2022, doi:10.3390/pharmaceutics14102151_

Round 1
Reviewer 1 Report
Kiong et al. developed an experimental group A Streptococcus vaccine based on synthetic epitopes in nanoparticles. The antigen consists of a lipopeptide (lipid core peptide or LCP) containing a B-cell epitope (J8), a T-helper epitope (P25). The core peptide is cationic and is complexed with anionic alginate to form nanoparticles. The LCP-alginate complex is stabilized by a layer of cationic polypeptide that is cross-linked by cysteine oxidation. The polypeptides were functionalized by adding potentially immune modulating functions: poly-Arg, poly-Arg-J8, mannosylated poly-Arg and poly-Lys. Particles were prepared and characterized with respect to encapsulation efficiency of LCP, particle size, charge and shape (EM). Immunization experiments were done in mice and antigen-specific antibody titers were measured as well as functional aspect of opsonic activity of the antibodies. Complete Freund’s adjuvanted LCP-1(?) was used as positive control.
Cationic nanoparticles were produced, often with low polydispersity. The nanoparticles induce substantial anti-J8 antibody responses at comparable levels, irrespective of the nanoparticle coat. Without coat (antigen-alginate) the responses are almost absent and the CFA control gives the highest responses. The opsonic activities of the induced antibodies shows clearer differences between the different nanoparticles. PEC-4 (cross linked poly-Lys coat) induces responses comparable to CFA adjuvanted antigen.
This is a small but interesting study but I have a number of comments and questions.
11. Abstract. ‘Peptide-based subunit vaccines include only minimal antigenic determinants and, therefore, are less likely to induce allergic immune responses and adverse effects compared to traditional vaccines.’ That is true but an epitope-based vaccine increases the chance of escape mutants and the immune response may be less complete.
22. Introduction. ‘However, they can present a variety of adverse effects as they carry a wide range of ‘redundant’ materials that do not contribute to achieving required immune responses.’ A considerable part of the adverse effects is caused by strong innate immune activation. This is not redundant and not only contributes to their adverse effects but also to excellent efficacy.
33. ‘For example, lipid core peptide (LCP) was able to induce strong humoral immune responses in conjunction with a variety of antigens without any adverse effects’. In mice? How was this measured? Animals are hardly predictive for safety (and efficacy for that matter) of vaccines in man. Suggest to delete claim.
44. M&M Group size is 5 mice. This is rather small so the study has limited statistical power. This may lead to incorrect conclusions (see later remark on incorrect interpretation of data). The claim that PEC-4 is as immunogenic as the CFA control may not be true.
55. The positive control, P25-J8/CFA, is not entirely clear to me. Is P25-J8 the same as LCP1? Use consistent nomenclature.
66. The design of animal study 2. A plain antigen control is missing, i.e. LCP-1. Instead, LCP-1 complexed to alginate is tested. This has opposite charge to all other preparations. Plain LCP-1 is cationic so the same charge as the nanoparticles and may have been a better control. What is the reason for omitting this?
77. Results. The encapsulation efficiency of LPC is determined but not reported? Please provide these data. In table 1 a mass ratio is given. Is this the initial ratio or actual measured values after manufacturing?
99. Figure 2 caption. ‘…comparing PEC-0 to PEC-4…’ should be PEC-1 to PEC-4?
110. Figure 2: add to p values: compared to P25-J8/CFA.
111. P9: ‘All PEC constructs induced antigen-specific immune responses at similar levels. PEC-4 induced the highest IgG titres of all formulations examined….’ This is contradictory. Responses cannot be similar and highest at the same time. A P-test relative to one of the PECs should indicate whether there are differences between the PEC groups or not. But cautioness is required because there are only 5 mice per group.
112. ‘It was notable that PEC-1 was able to trigger slightly higher IgG titres than PEC-2 and PEC-3.’ PEC-1 is not (significantly) higher. If that would be the case then the conclusion should also be that PEC-4 is lower than CFA. You cannot take statistics seriously for one comparison (PEC-2 is as immunogenic as CFA) and ignore it for another (PEC-1 is more immunogenic than PEC-3).
113. ‘This can be attributed to the presence of J8 epitope on the surface of PEC-1,..’ Why was J8 apparently not quantified? Is the J8 dose different between PEC-1 and the other PECs? Yes, according to the statement on P11. How much?
114. Supplementary: Figures S7 to S10: add sample codes (PEC-4, PEC-3, PEC-2, PEC-1) to the captions.
115. Fig. S11. There seems to be a discrepancy between DLS size (about 200 nm) and particle size on the micrographs (about 100 nm). Please comment.
116. Fig. S11. Only PEC-2 and PEC-4 are shown. Why not also the others? They are probably not as nice but such a bias should be prevented.
Reviewer 2 Report
Developing new vaccination strategies is important, thus the study "Development of Multilayer Nanoparticles for the Delivery of Peptide-based Subunit Vaccine against Group A Streptococcus" is actual and interesting. The experimental design is adequate and the study is within the scope of Pharmaceutics.
Only three minor comments:
1. Please decipher "solution A" and "solution B"
2. Was the sex of mice controlled?
3. Although poly-lysine is non-toxic, the toxicity of cross-linked poly-lysine should be studied.
